# The Diversity and Composition of Soil Microbial Communities Differ in Three Land Use Types of the Sanjiang Plain, Northeastern China

**DOI:** 10.3390/microorganisms12040780

**Published:** 2024-04-11

**Authors:** Shenzheng Wang, Mingyu Wang, Xin Gao, Wenqi Zhao, Puwen Miao, Yingnan Liu, Rongtao Zhang, Xin Wang, Xin Sui, Mai-He Li

**Affiliations:** 1Heilongjiang Provincial Key Laboratory of Ecological Restoration and Resource Utilization for Cold Region, Key Laboratory of Microbiology, Engineering Research Center of Agricultural Microbiology Technology, Ministry of Education, School of Life Sciences, College of Heilongjiang Province, Heilongjiang University, Harbin 150080, China; wangshenzheng2000@163.com (S.W.); wmy022234@163.com (M.W.); agaoxin0218@163.com (X.G.); puwenmiao@163.com (P.M.); tianronghaise@hlju.edu.cn (X.W.); 2Heilongjiang Provincial Natural Resources Rights and Interests Investigation and Monitoring Institute, Harbin 150088, China; kjydas2023@126.com; 3Institute of Nature and Ecology, Heilongjiang Academy of Sciences, Harbin 150001, China; liuyingn234@163.com (Y.L.); zhangrongtao14@163.com (R.Z.); 4Swiss Federal Institute for Forest, Snow and Landscape Research WSL, 8903 Birmensdorf, Switzerland; 5Key Laboratory of Geographical Processes and Ecological Security in Changbai Mountains, Ministry of Education, School of Geographical Sciences, Northeast Normal University, Changchun 130024, China; 6School of Life Science, Hebei University, Baoding 071002, China

**Keywords:** PLFA, soil microorganism, diversity, community composition

## Abstract

In recent years, the Sanjiang Plain has experienced drastic human activities, which have dramatically changed its ecological environment. Soil microorganisms can sensitively respond to changes in soil quality as well as ecosystem function. In this study, we investigated the changes in soil microbial community diversity and composition of three typical land use types (forest, wetland and cropland) in the Sanjiang Plain using phospholipid fatty acid analysis (PLFA) technology, and 114 different PLFA compounds were identified. The results showed that the soil physicochemical properties changed significantly (*p* < 0.05) among the different land use types; the microbial diversity and abundance in cropland soil were lower than those of the other two land use types. Soil pH, soil water content, total organic carbon and available nitrogen were the main soil physico-chemical properties driving the composition of the soil microbial community. Our results indicate that the soil microbial community response to the three different habitats is complex, and provide ideas for the mechanism by which land use changes in the Sanjiang Plain affect the structure of soil microbial communities, as well as a theoretical basis for the future management and sustainable use of the Sanjiang plain, in the northeast of China.

## 1. Introduction

Globally, ecosystems such as wetlands and forests are directly or indirectly threatened by human activities, including urbanization, agricultural expansion, timber harvesting, and climate change [1,2], which in turn lead to ecological destruction and degradation, and dysfunctional coupling of human–land systems. Land use change is the most direct reflection of human production activities, and the formation of its spatial pattern is the result of the joint action of natural and man-made factors. Many studies have shown that land use change can alter the environmental conditions of ecosystems and affect the quality and stability of ecosystems [3,4]. Additionally, the structure of soil microbial communities interacts with soil functions, can respond rapidly to changes in the soil environment, and is to some extent a sentinel for changes in environmental factors. Therefore, exploring the structure and diversity dynamics of soil microbial communities helps us to better understand the impacts of human activities (e.g., changes in management practices associated with different land use types) on the ecosystem.

Soil microorganisms play a crucial role in soil organic matter decomposition, energy flow, maintaining ecosystem function, and regulating global biogeochemical processes [5,6,7,8]. Changes in land use, such as wetland conversion to farmland, can significantly impact the composition and function of soil microbial community [9]. For example, Hua et al. [10] found that the relative abundance of Acidobacteria in plantation forests was significantly higher than that in agricultural fields, and the ratio of soil fungi to bacteria also changed significantly. Zhu et al. [11] reported that soil bacterial compositions and functions differed between wetlands and croplands under long-term human activity disturbance. The soil bacterial community composition was primarily affected by soil pH and soil texture, while the fungal community composition was most closely related to soil nutrient availability [11]. Liu et al. [12] revealed that land conversion directly impacted the expression of functional genes associated with assimilatory nitrate reduction in soil microorganisms. Therefore, the response of the soil microbial structure to different land use patterns is still uncertain due to different land use pattern and ecosystem types. We hypothesized that anthropogenic agricultural activities may disrupt the diversity and abundance of soil microbial communities in the Sanjiang Plain, potentially impacting regional ecosystem stability. Therefore, examining the influence of interactions between soil physicochemical properties and soil microbial community structure under wetland conversion could uncover new insights for managing the wetland ecosystem.

Located in the eastern part of Heilongjiang Province, in the middle of China’s largest concentration of freshwater wetlands, Sanjiang Plain is not only an important agricultural production base, but it also plays a very important role in protecting China’s food security [13]. It is rich in natural resources, and aside from the economic value it brings, there also exists an indispensable role in the protection of biodiversity, regulation of runoff, water purification, and the balance of regional nutrient cycling. The Sanjiang Plain acts as an important ecological barrier in Northeast Asia, which is of great significance to the global biochemical cycle as well as to the maintenance of ecological and climate security [14,15,16,17]. In addition to hydrology, topography, vegetation and other natural factors, the Sanjiang Plain has experienced drastic and excessive human activities (e.g., construction of artificial ditches, agricultural land reclamation, etc.) in the past few decades, which led to fragmentation of landscapes such as wetland forests and land degradation; thus, the ecological environment has changed dramatically [18,19]. Some studies claim that from the mid-20th century to the beginning of the 21st century, the area of wetlands decreased from 45.84% to 9.76%, the area of forests decreased from 29.39% to 25.34%, the area of arable land (including paddy fields and drylands) increased from 8.20% to 57.97%, and the area of grassland decreased from 15.36% to 4.65% [20]. The land structure, species diversity and ecosystem structure of the Sanjiang Plain have been altered with drastic human activities, which is why scientists have taken a keen interest in the study of the Sanjiang Plain in recent years. Some studies have shown that the deterioration of habitat conditions due to land use reduces fungal abundance but positively affects the abundance of certain fungi [21]. Sui et al. [22] found that microbial diversity and community structure significantly responded to land degradation successional gradients. Pellegrino et al. [23] found that land use patterns changes affect microbial communities in the long term, and they cannot recover their natural state even after many years. However, there is a gap in the research on the differences in microbial community structure among different land use types in the Sanjiang Plain. Therefore, it is important to study the effects of land use changes on soil microbial communities in the Sanjiang plain to help better understand future changes in wetland ecosystem functions due to human activities.

This study aims to investigate the effects of land use changes on soil physicochemical properties and the diversity and composition of soil microbial communities. We selected two typical land use types and an undisturbed control in the Sanjiang Plain that have been disturbed by human activities: natural wetlands, restored forests, and arable land. We investigated the diversity and composition of soil microbial communities using PFLA technology. Our specific objectives were (1) to investigate whether changes in land use types caused by human activities resulted in significant differences in microbial community structure, and (2) if so, whether soil physicochemical properties were the main cause of the differences in microbial community structure.

## 2. Materials and Methods

### 2.1. Study Site

The study site is located in Honghe National Nature Reserve of Sanjiang Plain, Northeast China (47°42′1800″–47°520′0700″ N, 133°340′3800″–133°460′2900″ E) (Figure 1). The study site exhibits a typical temperate humid/semi humid monsoon climate and exhibits an average annual temperature of 1.9 °C, with an average annual precipitation of 585 mm and evaporation of 1166 mm [22]. The soil in this region is classified as typical bleached stagnant soil and fibrous organic soil according to the USDA soil taxonomy. Three representative land use types were investigated in this study, including forest, wetland and cropland.

The wetland habitat was a pristine marsh meadow wetland that was seasonally flooded from May to August, and was the typical wetland type occurring in the Sanjiang Plain where *Deyeuxia angustifolia* is the dominant plant species. The arable land habitat was converted from a pristine marsh meadow wetland into a soybean plantation in 2007. The soybean plantation was fertilized with 370 kg ha^−1^ y^−1^ of fertilizer (N, P and K) each year in May. The forest habitat was plantations of Larix gmelinii that was planted on degraded wetlands in 2000. At the time of this study in 2016, the average height of the *Larix gmelinii* was about 7 m, the diameter at breast height was about 12 cm, and the average density was 1600 stems ha^−1^. No fertilization or forest management was conducted in this forest plantation. The wetland and arable land habitat types were approximately 400 m apart, and the forest plantation was located between them at a distance of approximately 200 m from each of the other two habitat types. The wetland, forest, and arable land covered a total area of 3000 m^2^, 2000 m^2^, and 1000 m^2^, respectively.

### 2.2. Soil Sampling Collecting

In July 2017, three 10 m × 10 m independent quadrats were set up in each land use type, with a distance of no less than 50 m between each quadrat. In each quadrat, ten soil samples were collected randomly from the soil surface (0–20 cm) using a 5 cm diameter soil auger, sieved (2 mm mesh) to remove sand, gravel, and coarse plant material, and then placed in ziplock bags stored in an ice box at 4 °C. Soil samples were then delivered immediately to the laboratory and divided into two parts; one was stored at −20 °C for microbiological analyses, and the other was dried for soil physico-chemical properties analysis.

### 2.3. Physico-Chemical Properties Analysis

Soil organic carbon (SOC) was determined using an elemental analyzer (VarioEL III, Langenselbold, Germany) [24]. Briefly, twenty-five mL of 0.3 mol/L KMnO_4_ solution was added to 0.5 g of soil, shaken for 30 min at room temperature, then centrifuged (5000 r/min) for 5 min, and the supernatant was diluted 250-fold with distilled water, and the absorbance at 565 nm was determined spectrophotometrically. A 1 mmol/L change in the concentration of KMnO_4_ corresponds to the oxidation of 9 mg of carbon. A pH analyzer (HQ30d; Hach Company, Loveland, CO, USA) was used to determine the pH value of soil. Soil moisture was determined via the drying method and determined gravimetrically by oven drying at 105 °C for 24 h; soil total nitrogen (TN) was determined with an elemental analyzer (Flash EA 1112 N, Thermo Fisher, Waltham, MA, USA); NO_3_^−^-N was determined via the phenol disulfonic acid colorimetric method; NH_4_^+^-N was determined via the potassium chloride leaching–indophenol blue colorimetric method. NH_4_^+^-N and NO_3_^−^-N were determined by the Elemental Analyzer (Flash EA 1112 N, Thermo Fisher, Waltham, MA, USA).

### 2.4. Phospholipid Fatty Acid (PLFA) Analysis

The PLFA analysis was used to determine soil microbial biomass and community composition [25,26]. We used a modified Bligh-Dyer method using an Agilent 5890 gas chromatograph (GC) (Santa Clara, CA, USA) with esterified C18:0 as an internal standard [27,28]. The chromatographic conditions were as follows: HP-5 column (30.0 µm × 320 µm × 0.25 µm); injection volume 1 µL; and carrier gas (N_2_) flow rate 0.8 mL/min. The initial temperature was 140 °C and held for 3 min, followed by an increase from 140 °C to 190 °C at a rate of 4 °C/min and held for 1 min, and finally an increase from 190 °C to 230 °C at a rate of 3 °C/min and held for 1 min. Detection was carried out using a flame ionization detector. BAME (Bacterial Acid Methyl Ester) mix and Supelco 37 Component FAME mix (Sigma-Aldrich, Burlington, MA, USA) were used for the identification and quantification of each fatty acid. Classification of microbiome groups based on detected PLFAs (as listed in Table 1) was based on the literature [29]. To facilitate the study of differences in soil microbial community composition and diversity, we further categorized the more abundant microbial species in the soil into different classes (e.g., actinomycetes are widely considered to be part of the Gram-positive bacteria, and the Gram-positive bacterial taxon numbering includes the actinomycetes) for separate comparisons.

### 2.5. Data Analysis

We organized the raw PLFA data such as total PLFA mass, composition, and content of each component expressed as a percentage of total PLFA. The individual characteristic PLFA biomarker amount was obtained by converting the reaction value of the internal standard 18:00 in ng/g with the following equation:$$N=\frac{\mathrm{Target}\;\mathrm{Response}}{(18:0)\mathrm{Response}}\times \left(18:0)\mathrm{Concentration}(\mathrm{ng}/g\right)\times \frac{\mathrm{Dissolved}\;\mathrm{sample}\;\mathrm{volume}(\mu L)}{\mathrm{Sample}\;\mathrm{dry}\;\mathrm{weight}(g)}$$
* where N is the amount of biomarker for the characteristic fatty acid type (ng/g); Response is the response value for the biomarker; 18:0 is the internal standard C18:0 (ng/µL); and the volume of the dissolved sample is in microliters, and the dry mass of the sample is in grams.

The number of various characteristic fatty acid biomarkers obtained by transformation was calculated, and the data were preliminarily processed using Excel 2016 (Microsoft, Redmond, WA, USA). The data were normalized for mean distribution before further analysis. One-way ANOVA and Duncan’s test were performed using SPSS19.0 (Released 2010. IBM SPSS Statistics for Windows, Version 19.0. Armonk, NY, USA: IBM Corp.) and the total number or ratio of bacterial, fungal, Gram-positive and Gram-negative markers were analyzed using Origin 2018 (OriginLab Corporation, Northampton, MA, USA). RDA analysis and heat maps were produced using R3.2 (Vegan package). The diversity index calculated as follows [30,31]:I.Shannon–Wiener (H): H = ∑(Pi)(lnPi);II.Simpson (D): D = 1 − Σpi2.

* Pi = Ni/N, where Ni is the number of characteristic fatty acids in treatment; n1 is the number of individuals with the first characteristic fatty acid biomarker; n2 is the number of individuals with the second characteristic fatty acid biomarker; n is the number.

Of individuals with the nth characteristic fatty acid biomarker.

III.Margalef (M): M = (S − 1)/lnN;IV.Menhinick (E): E = S/√N;V.Brillouin (B): B = N − 1 lg(N!/n1!n2!…n!).

* where N represents the number of total characteristic fatty acids in the experiment, and S is the total number of characteristic fatty acid markers.

## 3. Results

### 3.1. Changes in Soil Physico-Chemical Properties in Different Land Use Types

The soil physico-chemical properties (i.e pH, TN, TOC, NH_4_^+^-N, and SWC) changed significantly in three different land use types (wetland, forest and cropland), except for the content of soil NO_3_^−^-N (Table 2). The content of TN in soil was highest in cropland and the lowest in forest; the content of NH_4_^+^-N in cropland soil was significantly higher than that in wetland and forest, but there was no difference between wetland and forest; the contents of TOC and SWC in wetland soils were significantly higher than those of in forest and cropland; while the soil pH in forest was significantly higher than that in wetland and cropland.

### 3.2. Abundance of Soil Microbial PLFA in Different Land Use Types of the Sanjiang Plain

Seven soil microbial taxa (i.e., Gram-positive bacteria, Gram-negative bacteria, bacteria, actinomycetes, anaerobes, AM fungi) were dominant in all land use types (Figure 2). Moreover, the accumulative concentration of all PLFA biomarkers combined were wetland > forest > cropland.

Land use change significantly changed the abundance of soil bacteria, fungi, bacteria/fungi, GP, GN, GP/GN, actinomycetes and anaerobes (Figure 3). The abundance of soil bacteria was wetland > forest > cropland; the abundance of soil fungi was forest > wetland > cropland; the ratio of soil bacteria to soil fungi was cropland > wetland > forest; the abundance of GP and GN was wetland > forest > cropland, but the ratio of GP to GN was cropland > wetland > forest; the abundance of actinomycetes was wetland and forest > cropland; the abundance of anaerobes was wetland > forest > cropland.

### 3.3. Soil Microbial Alpha Diversity Indices of the Different Land Use Types

The soil microbial alpha indices (i.e., Shannon–Wiener, Simpson, Margalef, Menhinick, and Brillouin) were calculated based on PLFA values (Table 3). All soil microbial alpha indices of cropland were significantly lower than those of the two other land use types (Table 3, *p* < 0.05). The Shannon–Wiener, Simpson, Margalef and Brillouin indices showed that forest > wetland > cropland, while the Menhinick showed that forest = wetland > cropland (Table 3).

### 3.4. Relationship between Soil Microbial Structure and Soil Chemical Properties

Redundancy analysis was used to analyze the correlation with the soil microbial structure composition and the soil properties of the three land use types (Figure 4). The soil microbial community structure of wetland positively correlated with TOC (r^2^ = 0.918, *p* = 0.004), but negatively correlated with moisture content (r^2^ = 0.905, *p* = 0.001) (Figure 4). The soil microbial community structure of cropland, contrarily, positively correlated with moisture content, but negatively correlated with TOC (r^2^ = 0.905, *p* = 0.001). The soil microbial community structure of forest positively correlated with soil pH and NO_3_^−^-N.

A Venn diagram can be used to count the number of common and unique PLFAs in multiple samples and can intuitively show the compositional similarity and difference between environmental samples at different classification levels. As shown in Figure 5, the numbers of core PLFAs and unique PFLAs in the nine soil samples were 55 and 9, respectively. Among them, the forest soil sample contained a total of 61 PLFAs, and the number of unique PLFAs was 3. The total number of PLFAs in the wetland soil sample was 63, and the number of unique PLFAs was 4. The total number of PLFAs in the cropland soil sample was 60, and the unique PLFAs was 2.

We considered the correlation between soil physicochemical properties and the abundance of seven dominant microbial taxa based on the relative abundance profiles of microorganisms and recorded the results in a correlation heat map (Figure 6). Soil microorganisms showed different correlations with soil physicochemical properties. The most obvious difference in the heatmaps is the correlation between soil moisture content, which positively correlated with the abundance of soil actinomycetes, anaerobes, bacteria, Gram-positive bacteria and Gram-negative bacteria, while soil pH and NH_4_^+^N negatively correlated with the abundance of soil anaerobes, bacteria, Gram-positive bacteria and Gram-negative bacteria. TN was negatively correlated with fungi and actinomycetes, but not significantly correlated with most other microorganisms.

## 4. Discussion

### 4.1. Changes of Soil Physicochemical Properties under Different Land use Types in Sanjiang Plain

This study showed significant differences in soil physico-chemical properties between the three land use types (Table 2, *p* < 0.05). Anthropogenic fertilization in cropland and different natural hydrological conditions in forest and wetland may have contributed to this result. For example, TN in soil showed cropland > wetland > forest (Table 2), and in terms of water content, wetland was significantly higher than the other two land uses. The pH of wetland (5.61) and cropland (5.95) was slightly acidic, while forest (7.49) was slightly alkaline. TOC in wetland was significantly higher than in the other two land use types, while NH_4_^+^N was highest in cropland. The highest TOC content in the wetland may be due to the accumulation of organic matter over the years, but the abundant microorganisms in wetlands are unable to use these nutrient deposits [3], which is consistent with our findings. And the other two land use types have a longer air exposure time, which provides better conditions for organic carbon uptake and utilization by plants and microorganisms in them, in agreement with the findings of [32]. Our results showed that the TN content was highest in the cropland, which may be due to the application of fertilizer to increase the nitrogen content of the soil and the accumulation of crop residues by annual ploughing compared to the other two land use types, which allows for the accumulation of organic nitrogen content in the soil, and there are many researchers who have confirmed our view [33,34].

### 4.2. Effects of Different Land use Types on Soil Microbial Composition in Sanjiang Plain

We expressed microbial biomass in terms of PLFA quantities (ng/g). Bacterial biomass was highest in wetlands, and fungal biomass was highest in forests, but bacterial and fungal biomass were both lowest in cropland compared to forest and wetland (Figure 3a). Actinobacteria biomass of forest and wetland was similar, but significantly higher than that of cropland (Figure 3c). The biomass of Gram positive, Gram negative and anaerobes were all highest in wetland soil, while they were lowest in cropland (Figure 3b). In terms of differences in microbial biomass across land use types, the findings of Wang et al. [34] and Wang et al. [35] are consistent with our findings. However, other studies are contradictory to ours [3,36,37]. This may be related to soil moisture and soil nutrients. Hydrological conditions such as soil moisture and aeration could affect the abundance of microbial communities. Soil moisture content may affect microbial growth and metabolism, leading to changes in soil microbial communities [36,38]. Another reason may be the difference in the life cycles of the main fungi studied in this study, and the season of sampling and the weather (e.g., temperature and humidity) may also affect the structure of the microbial community as well as the physico-chemical properties of the soil, and this should be taken into account in future studies.

In this study, it was found that the main factors affecting the soil microbial community structure were different in the three land use types. Among them, TOC, water content, and NH_4_^+^N significantly affected soil microbial community structure in cropland and wetland soils, and pH significantly affected soil microbial community structure in forests. This is mainly due to the long-term high moisture content and anaerobic state of wetland soil, leading to slowly decomposing soil organic carbon and a high accumulation of organic carbon content, and cropland has a high nutrient content in the soil due to external fertilization. Many studies have shown that soil water content, TOC, and NH_4_^+^N affect the microbial community structure [39,40,41,42], which is in agreement with our findings. However, Clairmont et al. [39] found that the soil microbial community structure was also influenced by aboveground vegetation composition, and this should be taken into account in future studies of soil microbial communities.

### 4.3. Effects of Different Land Use Types on Soil Microbial Diversity

This results showed that soil microbial community composition and diversity changed significantly in the three different land use types. The α-diversity indices of forests and wetlands were significantly higher than those of cropland, which is consistent with the previous findings [3,37]. The environmental conditions of cropland soils are affected by disturbances such as agricultural tillage, and their soils are rich in nutrients but have a single above-ground vegetation type and low abundance of deadfall inputs, so the diversity of soil microorganisms in cropland is lower than that of the other two land use types. Lynn et al. [43] found that soil microbial diversity in wetlands is higher than that in cropland because land use interferes with ecological stability, which reduces the soil microbial diversity. Suleiman et al. [44] also found that deforestation leads to a decrease in soil microbial diversity, which is consistent with our findings.

The application of PLFA techniques to characterize soil microbial communities in different land use types has certain disadvantages: (1) Not all characteristic fatty acids of soil microbes are known, and, therefore, the specific fatty acids that can be detected cannot always be accurately associated with the soil microbes that produce them. (2) PLFA techniques rely on labeling fatty acids and determining the structure of the microbial community based on the location of the fatty acids. Changes in the location of the marker or inaccurate labeling can affect the results. (3) PLFA technology cannot detect changes in the metabolism and function of soil microbial communities due to changes in the environment. However, PLFA is complementary to other techniques (e.g., high-throughput sequencing and biology technologies).

## 5. Conclusions

Phospholipid fatty acid (PLFA) analysis was used to quantitatively determine the types and contents of soil microbial PLFA in different land use types in the Sanjiang Plain. The results showed that the soil microbial community structure varied significantly in three land use types, with wetlands having the highest soil microbial community diversity and cropland having the lowest soil microbial community diversity and microbial abundance. Soil pH, water content, TOC and available nitrogen content were the main environmental factors affecting soil microbial community composition. Changes in land use types caused by human activities can lead to significant differences in microbial community structure due to changes in soil physico-chemical properties. This result can be used to predict the effects of land use changes on soil microbial communities. Future studies should incorporate other techniques (e.g., high-throughput sequencing) to more deeply investigate the changes in soil microbial communities and functions due to land use changes.

## Figures and Tables

**Figure 1 microorganisms-12-00780-f001:**
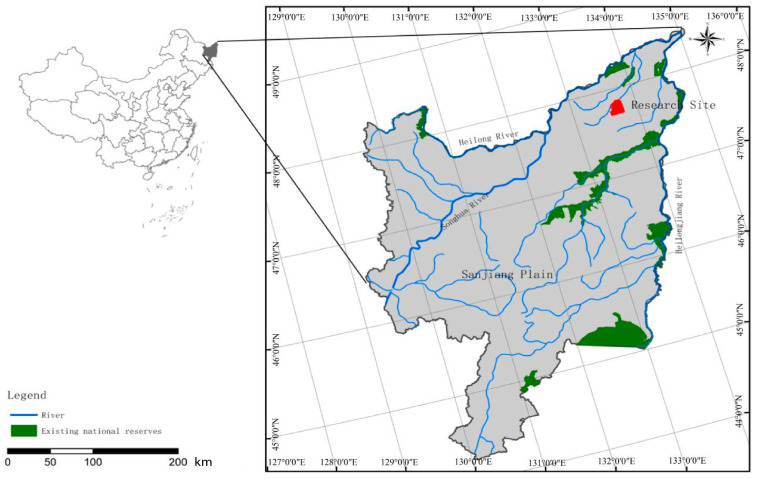
The location of this study. The red indicates this research site.

**Figure 2 microorganisms-12-00780-f002:**
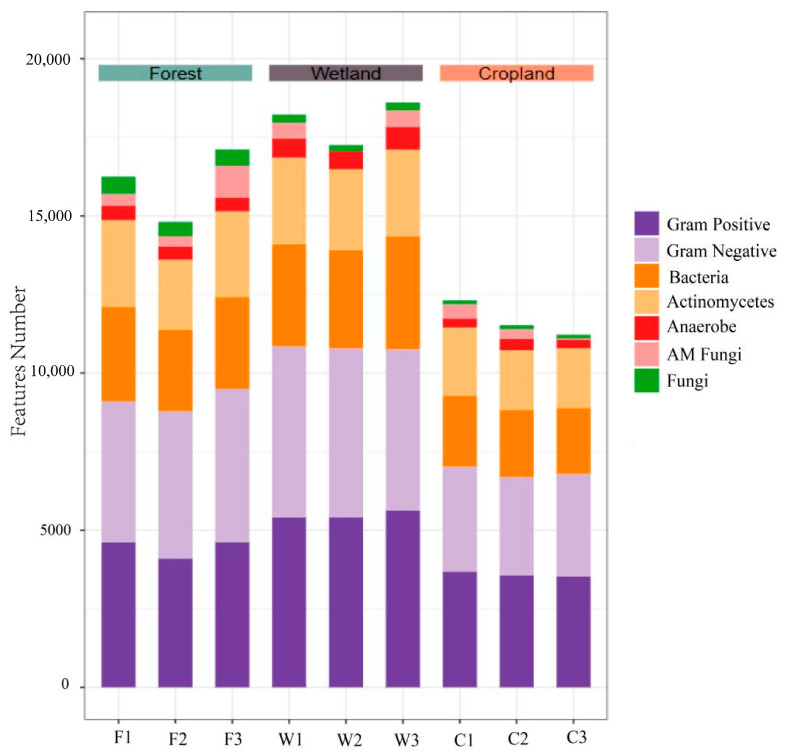
Stacked histograms of the number of different soil microbial PLFA markers in three different land use types of Sanjiang plain.

**Figure 3 microorganisms-12-00780-f003:**
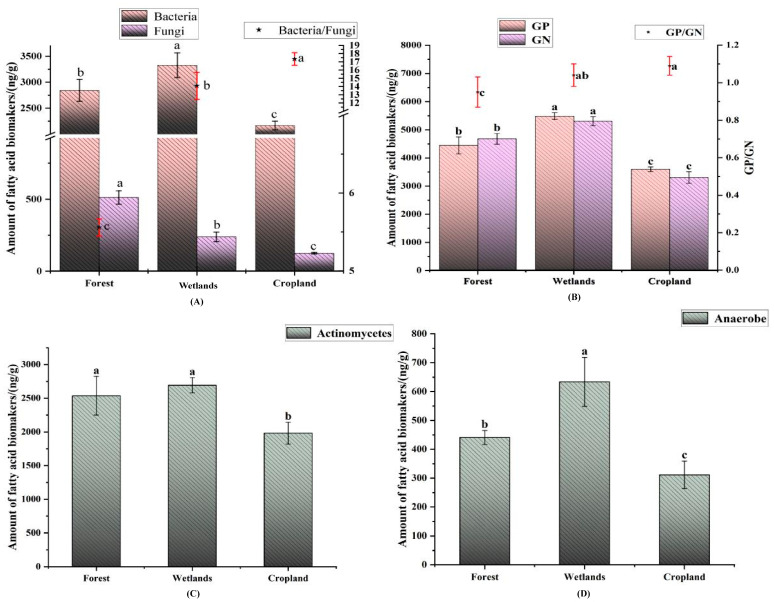
Totals of detected PLFAs (ng/g) of characteristic soil microbial communities from different wetlands in the Sanjiang Plain. (**A**): Accumulative PLFA biomarkers indicative of unspecified bacteria and fungi, respectively, and bacteria/fungi ratio. “★” represents the ratio of bacteria to fungi. (**B**): Accumulative PLFAs for Gram-positive bacteria (GP) and Gram-negative bacteria (GN) bacteria and their ratio.“★” represents the ratio of Gram-positive bacteria to Gram-negative bacteria. (**C**): Accumulative PLFAs levels indicative of actinomycetes, and (**D**) of anaerobes. The different letters (a–c) represent significant differences between treatments.

**Figure 4 microorganisms-12-00780-f004:**
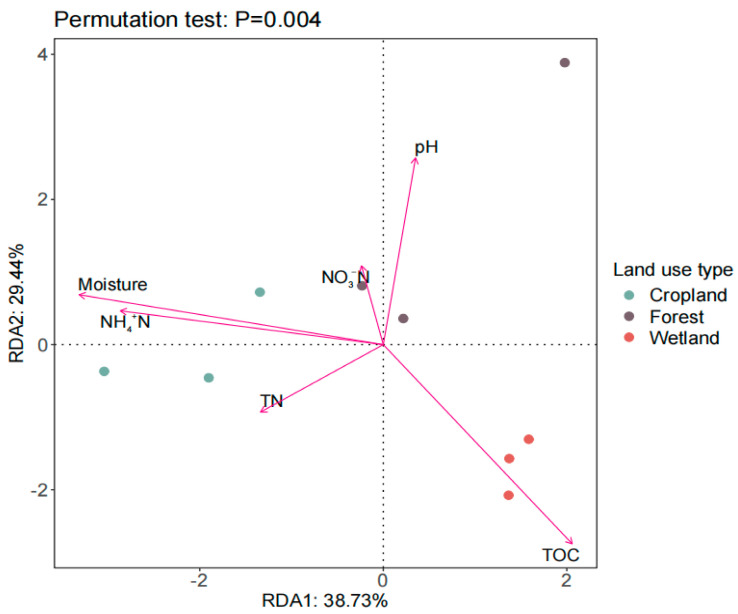
Redundancy analysis (RDA) conducted with the soil microbial PLFA-derived data and soil physicochemical properties of the three different land use types in the Sanjiang Plain. TOC: total organic carbon; TN: total nitrogen; NO_3_^−^-N, nitrate nitrogen.

**Figure 5 microorganisms-12-00780-f005:**
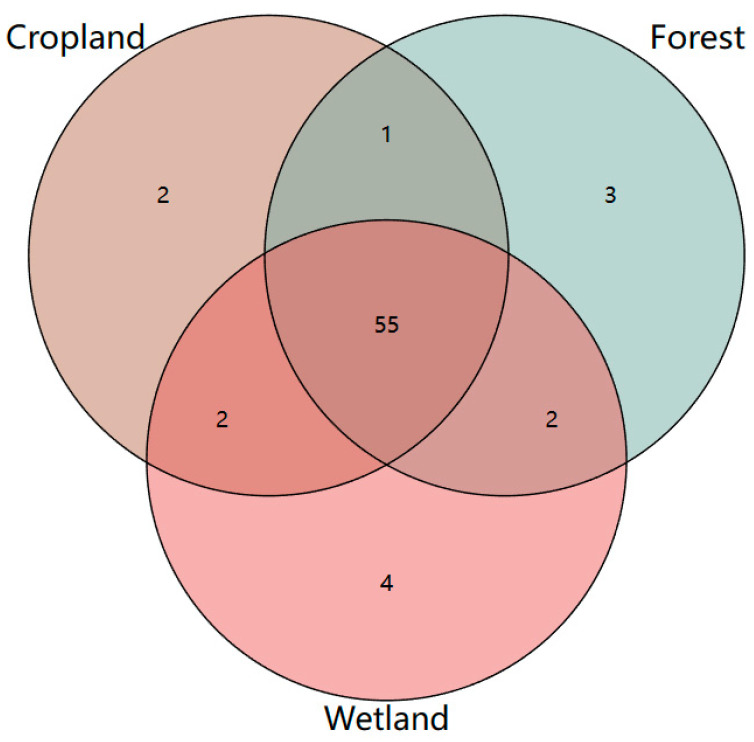
Comparison of the number of PLFAs in different land use types.

**Figure 6 microorganisms-12-00780-f006:**
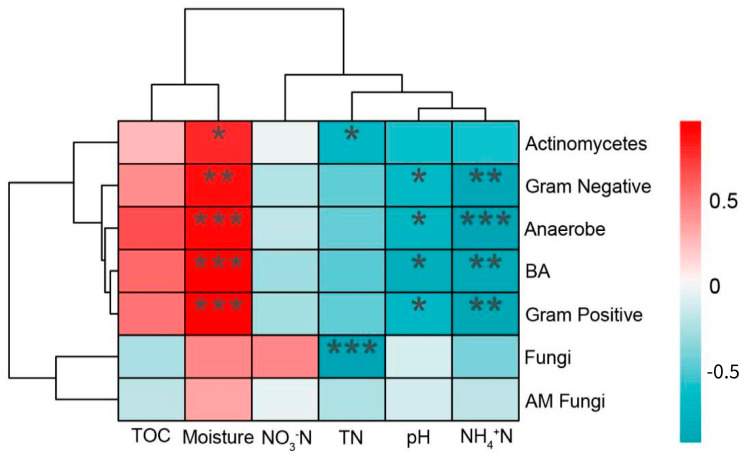
Heat map showing correlations between soil microbial PLFA markers and soil physicochemical properties in the three land use types. Statistical significance is given as * *p* < 0.05, ** *p* < 0.01, *** *p* < 0.001. TOC: total organic carbon; TN: total nitrogen.

**Table 1 microorganisms-12-00780-t001:** The phospholipid fatty acid (PLFA) markers used to identify particular soil microbial community groups.

Microbial Group	Peak Name
Bacteria	12:0, 13:0, 14:0, 15:0, 16:0, 18:0, 22:0, 24:0
Gram-Positive	11:0 iso, 11:0 anteiso, 12:0 iso, 12:0 anteiso, 13:0 iso, 13:0 anteiso, 14:1 iso ω7c, 14:0 iso, 14:0 anteiso, 15:1 iso ω9c, 15:1 iso ω6c, 15:1 anteiso ω9c, 15:0 iso, 15:0 anteiso, 16:0 iso, 16:0 anteiso, 17:1 iso ω9c, 17:0 iso, 17:0 anteiso, 18:0 iso, 19:0 iso, 19:0 anteiso, 20:0 iso, 22:0 is0
Gram-Negative	13:1 ω5c, 13:1 ω4c, 13:1 ω3c, 12:0 2OH, 14:1 ω7c, 14:1 ω5c 15:1 ω9c, 15:1 ω8c, 15:1 ω7c, 15:1 ω6c, 15:1 ω5c, 14:0 2OH, 16:1 ω9c, 16:1 ω7c 16:1 ω6c, 16:1 ω4c, 16:1 ω3c, 17:1 ω9c, 17:1 ω8c, 17:1 ω7c, 17:1 ω6c, 17:0 cyclo ω7c, 17:1 ω5c, 17:1 ω4c, 17:1 ω3c, 16:0 2OH, 18:1 ω8c, 18:1 ω7c, 18:1 ω6c, 18:1 ω5c, 18:1 ω3c, 19:1 ω9c, 19:1 ω8c, 19:1 ω7c, 19:1 ω6c, 19:0cyclo ω9c, 19:0 cyclo ω7c, 19:0 cyclo ω6c, 20:1 ω9c, 20:1 ω8c, 20:1 ω6c, 20:1 ω4c, 20:0 cyclo ω6c, 21:1 ω9c, 21:1 ω8c, 21:1 ω6c, 21:1 ω5c, 21:1 ω4c, 21:1 ω3c, 22:1 ω9c, 22:1 ω8c, 22:1 ω6c 22:1 ω5c, 22:1 ω3c, 22:0 cyclo ω6c, 24:1 ω9c, 24:1 w7c
Actinomycetes	16:0 10-methyl, 17:1 ω7c 10-methyl, 17:0 10-methyl, 18:1 ω7c10-methyl, 18:0 10-methyl, 19:1 ω7c 10-methyl, 20:0 10-methyl
Fungi	18:1ω9c, 18:2 ω6c, 23:0
Protozoa	19:3ω6c, 20:3ω6c, 20:4ω6c, 20:5ω3c
AM Fungi	16:1 ω5c
Anaerobic	13:0 DMA, 15:0 DMA, 16:2 DMA, 16:1 w9c DMA, 16:1 w7c DMA, 17:0 DMA, 18:2 DMA, 18:1 w9c DMA, 18:1 w7c DMA, 18:0 DMA

The classification of the microbiome was based on the literature [29].

**Table 2 microorganisms-12-00780-t002:** Soil physicochemical properties of the different land use types.

Site	pH	TN (g/kg)	TOC (g/kg)	NH_4_^+^N (mg/kg)	NO_3_^−^N (mg/kg)	SWC (%)
Forest	7.49 ± 0.35 a	1.89 ± 0.40 c	19.22 ± 1.99 b	16.85 ± 1.83 b	0.27 ± 0.05 a	0.23 ± 0.00 b
Wetland	5.61 ± 0.37 b	3.56 ± 0.39 b	40.63 ± 1.03 a	10.65 ± 0.92 b	0.15 ± 0.02 a	0.43 ± 0.13 a
Cropland	5.95 ± 0.12 b	4.63 ± 0.56 a	20.34 ± 1.67 b	55.53 ± 5.62 a	0.22 ± 0.12 a	0.16 ± 0.02 b

Values are given as mean ± standard error (*n* = 3); different letters represent significant differences between treatments (*p* < 0.05). TN: total nitrogen; TOC: total organic carbon; SWC: soil water content. NH_4_^+^N: ammonium nitrogen; NO_3_^−^N: Nitrate nitrogen.

**Table 3 microorganisms-12-00780-t003:** Soil Microbial alpha diversity indices of the three different land use types, based on PLFA data.

Diversity Index	Forest	Wetland	Cropland
Shannon–Wiener (H)	3.29 ± 0.07 a	3.21 ± 0.02 a	2.23 ± 0.08 b
Simpson (D)	0.95 ± 0.01 a	0.94 ± 0.00 a	0.78 ± 0.07 b
Margalef (M)	4.68 ± 0.11 a	4.64 ± 0.23 a	3.34 ± 0.09 b
Menhinick (E)	0.35 ± 0.01 a	0.35 ± 0.02 a	0.28 ± 0.04 b
Brillouin (B)	3.23 ± 0.05 a	3.12 ± 0.06 a	2.65 ± 0.05 b

Different letters in the same row indicate significant differences between different land use types Note: different lowercase letters (a,b) represent comparisons at the 0.05 significance level.

## Data Availability

Data are contained within the article.

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
