# Peer review of "The Diversity and Composition of Soil Microbial Communities Differ in Three Land Use Types of the Sanjiang Plain, Northeastern China"

_microorganisms, 2024, doi:10.3390/microorganisms12040780_

Round 1

Reviewer 1 Report

Comments and Suggestions for Authors

I have revised the manuscript microorganisms-2928620. This is a very interesting study that aims to investigate the changes in soil microbial community diversity and composition across three typical land use types (forest, wetland, and cropland) in the Sanjiang Plain using phospholipid fatty acid analysis (PLFA) technology. The use of PLFA is robust and provides a deeper understanding of microbial community composition.

Regarding manuscript ID microorganisms-2928620, specific questions are addressed as follows:

What is the main question addressed by the research?

The authors aim to investigate the effects of land use changes on soil physicochemical properties and the diversity and composition of soil microbial communities. However, I can not find the main hypothesis that they tested. The authors must clarify the main hypothesis of the manuscript in the introduction section.

What parts do you consider original or relevant for the field?

The results and discussion contribute original and relevant insights to the field of soil microbiology and the use of PLFA technology.

What specific gap in the field does the paper address?

The results of this study can be used to predict the effects of land use changes on soil microbial communities. Future studies should incorporate other techniques (e.g., high-throughput sequencing) or other abiotic stressors to more deeply investigate the changes in soil microbial communities and functions due to land use changes. 

What does it add to the subject area compared with other published material?

The paper provides an interesting dataset about soil microbiota community composition by using PLFA technology.

What specific improvements should the authors consider regarding the methodology? What further controls should be considered?

The experimental design is robust, and no further controls are necessary. However, the authors must reduce the amount of wording duplication in the entire manuscript.

Regarding the consistency of conclusions with evidence and arguments presented, all main questions were addressed in the results and discussion sections, and the conclusions align with the study's aim. However, I reiterate there are a lot of word duplication. The authors must revise and rewrite some sentences. See the iThenticate report.

Some references are not appropriate. The authors must update references older than 6 years.

Tables and figures adhere to the author's guidelines, with high-resolution figures and informative tables, but Figure 2 must have its resolution improved.

Author Response

Dear Editor Dr. YANG Qin,

Dear Reviewer 1,

Thank you for your letter and for the comments concerning our manuscript entitled “The Diversity and Composition of Soil Microbial Community Differ in Three Land use Types of the Sanjiang Plain, Northeast-ern China (microorganisms-2928620)”. Those comments are all valuable and very helpful for revising and improving our paper. We have studied all provided comments carefully and have made appropriate corrections which we hope meet with approval. The corrections made in the paper and the respective responses to your comments are listed below and shown by revision format in the improved version of the text.

Comments to the author (if any):

Reviewer #1: I have revised the manuscript microorganisms-2928620. This is a very interesting study that aims to investigate the changes in soil microbial community diversity and composition across three typical land use types (forest, wetland, and cropland) in the Sanjiang Plain using phospholipid fatty acid analysis (PLFA) technology. The use of PLFA is robust and provides a deeper understanding of microbial community composition. Correct the following observations:

Q1.  What is the main question addressed by the research?

The authors aim to investigate the effects of land use changes on soil physicochemical properties and the diversity and composition of soil microbial communities. However, I can not find the main hypothesis that they tested. The authors must clarify the main hypothesis of the manuscript in the introduction section.

Response:

Dear reviewer,

Thank you for your constructive comment for our manuscript. We agree with you and revise the introduction according to your suggestion.

Please see the chapter of introduction.

“It is hypothesized that anthropogenic agricultural activities may disrupt the diversity and abundance of soil microbial communi-ties in the Sanjiang Plain, potentially impacting regional ecosystem stability.” “Our specific objectives were (1) to investigate whether changes in land use types caused by human activities resulted in signifi-cant differences in microbial community structure, and (2) if so, whether soil physicochemical properties were the main cause of the differences in microbial community structure.”

Q2.  What parts do you consider original or relevant for the field?

The results and discussion contribute original and relevant insights to the field of soil microbiology and the use of PLFA technology.

Response:

Dear reviewer, thank you for your comments, we will combine other methods in our next research to study the changes in soil microbial communities and functions due to land use changes in more depth.

Q3.  What specific gap in the field does the paper address?

The results of this study can be used to predict the effects of land use changes on soil microbial communities. Future studies should incorporate other techniques (e.g., high-throughput sequencing) or other abiotic stressors to more deeply investigate the changes in soil microbial communities and functions due to land use changes.

Response:

Thank you for your constructive comments. We agree with your comments and have revised the conclusions based on your comments. Please see the revised format in the Conclusions chapter.

"Our results suggest that anthropogenic changes in land use types can lead to significant differences in microbial community structure, while changes in soil physical and chemical properties are the main cause of differences in microbial community structure. "

Q4.  What does it add to the subject area compared with other published material?

The paper provides an interesting dataset about soil microbiota community composition by using PLFA technology.

Response:

Thank you for your comments, and our future research will incorporate other techniques (e.g., high-throughput sequencing) or other abiotic stressors to more deeply investigate changes in soil microbial communities and function due to land use change.

Q5.  The authors must reduce the amount of wording duplication in the entire manuscript.

Response:

Thank you for your constructive comments on our manuscript. We agree with your comments and have modified the wording to reduce repetitive descriptions in accordance with your suggestions.

Q6.  The authors must update references older than 6 years.

Response:

Thank you for your constructive comments on our manuscript. We agree with your comments and have updated the references according to your suggestions.

Reviewer 2 Report

Comments and Suggestions for Authors

The paper "The Diversity and Composition of Soil Microbial Community Differ in Three Land use Types of the Sanjiang Plain, Northeastern China "is highly cognitively charged bringing new insights into bacterial identification-but it does not fully clearly present its methods and results. Here are my most important suggestions and comments:

Table 2. No literature sources available for Table 2.

Table 3.  Lacks units for NH4+N and NO3-N in table 3.

No literature sources for how to calculate the Shannon-Wiener and Simpson index.

There is too little justification in the paper for PLFA in relation to 16srRNA sequence analysis as a method of bacterial identification - no literature citations in this area.

Author Response

Dear Editor Dr. YANG Qin,

Dear Reviewer 1, 2 and 3,

Thank you for your letter and for the comments concerning our manuscript entitled “The Diversity and Composition of Soil Microbial Community Differ in Three Land use Types of the Sanjiang Plain, Northeast-ern China (microorganisms-2928620)”. Those comments are all valuable and very helpful for revising and improving our paper. We have studied all provided comments carefully and have made appropriate corrections which we hope meet with approval. The corrections made in the paper and the respective responses to your comments are listed below and shown by revision format in the improved version of the text.

Reviewer #2: Dear editor

The paper "The Diversity and Composition of Soil Microbial Community Differ in Three Land use Types of the Sanjiang Plain, Northeastern China "is highly cognitively charged bringing new insights into bacterial identification-but it does not fully clearly present its methods and results. Here are my most important suggestions and comments:

Major Comments:

Q1.  Table 2. No literature sources available for Table 2.

Response:

Dear reviewer,

Thank you for your constructive comment for our manuscript. We agree with you and revise the Table 2 according to your suggestion.

Please see the chapter of table 2.

Q2.  Table 3.  Lacks units for NH4+N and NO3-N in table 3.

Response:

Dear reviewer,

Thank you for your constructive comment for our manuscript. We agree with you and revise the table 3 according to your suggestion. Please see the table 3.

Q3.  No literature sources for how to calculate the Shannon-Wiener and Simpson index.

Response:

Thank you for your constructive comments. We agree with your comments and have added references based on your comments. Please refer to 2.5. in the Material Methods chapter.

Q4.  There is too little justification in the paper for PLFA in relation to 16srRNA sequence analysis as a method of bacterial identification - no literature citations in this area.

Response:

Thank you for your constructive comments. We agree with your comments and have added references based on your comments. Please refer to 2.4. in the Material Methods chapter.

Reviewer 3 Report

Comments and Suggestions for Authors

The manuscript by Wang et al. outlines a comparison of microbiota in a wetland versus two examples of wetland turned to cultivation. The authors used PFLA to quantify and characterize the microbiota. In line with many other studies, their results point to loss of microbial diversity following switch of natural soil system to monoculture. The manuscript is well written, but is lacking in some key pieces of information, outlined below.

The majority of microbiome studies over recent decades have used sequencing of partial 16SrRNA gene amplicons. While this is not the only suitable approach to obtain microbiome information, the many findings based thereon cannot be ignored. The authors should add a substantial introductory section that outlines how PFLA can be used to characterize microbiomes, and how its results compare to those derived from 16SrRNA gene amplicon data. Clearly both approaches have strengths and weaknesses.

The allocation of PFLA data to taxa is confusing to me as the groups seem to overlap. Actinomycetes are widely viewed as part of Gram positive, so would the Gram-positive group numbers include actinomycetes or not? Likewise, “Anaerobic” can include Gram positive and Gram negatives. Please clarify this throughout the manuscript. Giving pertinent background on this in the introduction is important – see avove.

Specific points:

1.     Intro para 1 line 5: What is meant by “intuitive” here?

2.     Intro para 1 line 9: Change “addition” to “additionally”. Change “interacts with” to “is an integral part of “

3.     Intro para 3 line 3rd line from end: If you state that something is unclear, it means there is some info but not enough. If, as you indicate, there is no information, then write “no information”.

4.     Intro para 4: You say that you study three different land use types, but was it not two land use types and an undisturbed control?

5.     Methods para 2 line 3: Deyeuxia angustifolia in italics. Please check the entire manuscript to ensure all genus and species names are italicized.

6.     Methods para 2: “N:P:K” – define ratios

7.     Figure 1: The latitude and longitude numbers are hard to read – can you enlarge please?

8.     Methods 2.4 line 2: Briefly outline how fatty acids were extracted. Give a reference for the Bligh-Dyer method.

9.     Table 2 should be Table 1, and then Table 3 should be 2 – please also correct in the text.

10.  Table 2 header: Please state who defined which PFLA markers to assign to which taxa – some reference please.

11.  Figure 3A: This is complex, and I was not able to figure out what the numbers on top of the bars meant. They do not seem to fit with the scale on the Y-axis.

12.  Discussion 42: line 1: “labelling” – do you mean “quantities”?

13.  Page 11 line 1: In what specific ways do those studies contrast with yours?

14.  Page 11 line 4: Should “colonization: be “growth”?

15.  Page 11 line 6: What is meant by “life cycles” here. Fungo tend to have these, but most bacteria do not have a true life cycle (OK, B, C and D phase, but that is just part of growth).

16.  Discussion 4.3 line 1: “…microbial community composition and diversity…”

Author Response

Dear Editor Dr. YANG Qin,

Dear Reviewer3,

Thank you for your letter and for the comments concerning our manuscript entitled “The Diversity and Composition of Soil Microbial Community Differ in Three Land use Types of the Sanjiang Plain, Northeast-ern China (microorganisms-2928620)”. Those comments are all valuable and very helpful for revising and improving our paper. We have studied all provided comments carefully and have made appropriate corrections which we hope meet with approval. The corrections made in the paper and the respective responses to your comments are listed below and shown by revision format in the improved version of the text.

Reviewer #3: Dear editor

The manuscript by Wang et al. outlines a comparison of microbiota in a wetland versus two examples of wetland turned to cultivation. The authors used PFLA to quantify and characterize the microbiota. In line with many other studies, their results point to loss of microbial diversity following switch of natural soil system to monoculture. The manuscript is well written, but is lacking in some key pieces of information, outlined below.

The majority of microbiome studies over recent decades have used sequencing of partial 16SrRNA gene amplicons. While this is not the only suitable approach to obtain microbiome information, the many findings based thereon cannot be ignored. The authors should add a substantial introductory section that outlines how PFLA can be used to characterize microbiomes, and how its results compare to those derived from 16SrRNA gene amplicon data. Clearly both approaches have strengths and weaknesses.

The allocation of PFLA data to taxa is confusing to me as the groups seem to overlap. Actinomycetes are widely viewed as part of Gram positive, so would the Gram-positive group numbers include actinomycetes or not? Likewise, “Anaerobic” can include Gram positive and Gram negatives. Please clarify this throughout the manuscript. Giving pertinent background on this in the introduction is important – see avove.

Here are my most important suggestions and comments:

Major Suggestions:

Q1.  The authors should add a substantial introductory section that outlines how PFLA can be used to characterize microbiomes, and how its results compare to those derived from 16SrRNA gene amplicon data.

Response:

Dear reviewer,

Thank you for your comments, at the end of Discussion 4.3. we presented the drawbacks of the PLFA technique, and our future research will look more deeply into changes in soil microbial communities and functions due to land use change in combination with other techniques (e.g., high-throughput sequencing) or other abiotic stressors.

Q2.  The allocation of PFLA data to taxa is confusing to me as the groups seem to overlap. Actinomycetes are widely viewed as part of Gram positive, so would the Gram-positive group numbers include actinomycetes or not? Likewise, “Anaerobic” can include Gram positive and Gram negatives. Please clarify this throughout the manuscript. Giving pertinent background on this in the introduction is important – see avove.

Response:

Dear reviewer,

Thank you for your constructive comments on our manuscript. We agree with your comments and have added more detailed formulations based on your suggestions. Please refer to the end of the Materials and Methods section 2.4.

Major Comments:

Q1. Intro para 1 line 5: What is meant by “intuitive” here?

Response:

Dear reviewer,

Thank you for your constructive comment for our manuscript. We agree with your comments and have revised the presentation based on your suggestions. Please refer to Intro para 1 line 5.

Q2. Intro para 1 line 9: Change “addition” to “additionally”. Change “interacts with” to “is an integral part of “

Response:

Thank you for your constructive comments on our manuscript. We agree with your comments and have revised it according to your suggestions. Please see the Intro para 1 line 9.

Q3. Intro para 3 line 3rd line from end: If you state that something is unclear, it means there is some info but not enough. If, as you indicate, there is no information, then write “no information”.

Response:

Thank you for your constructive comments. We agree with your comments and have modified the sentence accordingly. Please refer to Intro para 3 line 3rd line from end.

Q4. Intro para 4: You say that you study three different land use types, but was it not two land use types and an undisturbed control?

Response:

Thank you for your constructive comments on our manuscript. We agree with your comments and have revised the sentences to use more accurate descriptions as you suggested.

Q5. Methods para 2 line 3: Deyeuxia angustifolia in italics. Please check the entire manuscript to ensure all genus and species names are italicized.

Response:

Thank you for your constructive comments on our manuscript. We agree with your comments and have checked and changed the fonts according to your suggestions.

Q6. Methods para 2: “N:P:K” – define ratios

Response:

We agree with your comments and have changed the description according to your suggestions.

Q7. Figure 1: The latitude and longitude numbers are hard to read – can you enlarge please?

Response:

Thank you for your constructive comments on our manuscript. We agree with your comments and have revised the Figure 1.

Q8. Methods 2.4 line 2: Briefly outline how fatty acids were extracted. Give a reference for the Bligh-Dyer method.

Response:

Thank you for your constructive comments on our manuscript. We agree with your comments and have added references based on your suggestions.

Q9. Table 2 should be Table 1, and then Table 3 should be 2 – please also correct in the text.

Response:

Thank you for your constructive comments on our manuscript. We agree with your comments and have changed the description according to your suggestions.

Q10. Table 2 header: Please state who defined which PFLA markers to assign to which taxa – some reference please.

Response:

Thank you for your constructive comments on our manuscript. We agree with your comments and have added references based on your suggestions.

Q11. Figure 3A: This is complex, and I was not able to figure out what the numbers on top of the bars meant. They do not seem to fit with the scale on the Y-axis.

Response:

Thank you for your constructive comments on our manuscript. We agree with your comments and have modified Figure 3a according to your suggestions.

Q12. Discussion 42: line 1: “labelling” – do you mean “quantities”?

Response:

Thank you for your constructive comments on our manuscript. We agree with your comments and have revised the description according to your suggestions.

Q13. Page 11 line 1: In what specific ways do those studies contrast with yours?

Response:

Thank you for your constructive comments on our manuscript. We agree with your comments and have used a more detailed description based on your suggestions.

Q14. Page 11 line 4: Should “colonization: be “growth”?

Response:

Thank you for your constructive comments on our manuscript. We agree with your comments and have changed the description according to your suggestions.

Q15. Page 11 line 6: What is meant by “life cycles” here. Fungo tend to have these, but most bacteria do not have a true life cycle (OK, B, C and D phase, but that is just part of growth).

Response:

Thank you for your constructive comments on our manuscript. We agree with your comments and have changed the description according to your suggestions.

Q16. Discussion 4.3 line 1: “…microbial community composition and diversity…”

Response:

Thank you for your constructive comments on our manuscript. We agree with your comments and have revised the description according to your suggestions.

Round 2

Reviewer 3 Report

Comments and Suggestions for Authors

I am disappointed that you did not add any information on PFLA versus 16SrRNA gene sequencing in the introduction as I had requested.